# Antioxidant and Antimicrobial Activity of Ferulic Acid Added to Dried Meat: Shelf-Life Evaluation

**DOI:** 10.3390/foods14040708

**Published:** 2025-02-19

**Authors:** Any Guadalupe Hernández-Jaime, Francisco Castillo-Rangel, Martha María Arévalos-Sánchez, Ana Luisa Rentería-Monterrubio, Eduardo Santellano-Estrada, Juan Manuel Tirado-Gallegos, América Chávez-Martínez

**Affiliations:** Facultad de Zootécnica y Ecología, Universidad Autónoma de Chihuahua, Periférico Francisco R. Almada, Km 1, Chihuahua 31453, Mexico; p310330@uach.mx (A.G.H.-J.); fcastillor@uach.mx (F.C.-R.); marevalos@uach.mx (M.M.A.-S.); arenteria@uach.mx (A.L.R.-M.); esantellano@uach.mx (E.S.-E.); jtirado@uach.mx (J.M.T.-G.)

**Keywords:** ferulic acid, nitrites, dried meat, antioxidant activity, lipid oxidation

## Abstract

Ferulic acid is an antimicrobial and antioxidant phenolic compound located in the cell walls of plants and therefore classified as a natural antioxidant. The objective of this study was to assess the antimicrobial and antioxidant potential of ferulic acid as a substitute for nitrites in the elaboration of dried meat. Four treatments were evaluated: dried meat without nitrites or ferulic acid, (control treatment), dried meat with nitrites, dried meat with 0.05% of ferulic acid, and dried meat with 0.1% of ferulic acid. The antioxidant activity, lipid oxidation, and microbiological quality were evaluated throughout the dried meat shelf life. The protein, fat, and ash content was not different between the treatments with nitrites and ferulic acid (*p* > 0.05) and all values were within the ranges established for these nutrients. Regarding the moisture content, although there was a difference between treatments (*p* < 0.05), the values found were within the reported range (5–15%) in dried meat. Treatment with nitrites had the highest sodium content (*p* < 0.05), although all treatments surpass the daily consumption of sodium recommended by the World Health Organization. In addition, color differences would not be noticeable to the human eye. Treatments with ferulic acid exhibited the highest (*p* < 0.05) antioxidant activity and the lowest lipid oxidation and total aerobic mesophile counts. Finally, the change in the formulation of dried meat using ferulic acid instead of nitrites was not perceptible to panelists in sensory evaluation. These findings suggest that the incorporation of ferulic acid, when added to dried meat, can improve its oxidative stability and increase its antioxidant activity. In conclusion, the use of ferulic acid at a concentration of 0.1% is recommended because, at this concentration, the antioxidant activity was greater, and the oxidation was below the threshold of perceived rancidity. However, further research is needed to study the effect of nitrite substitution using ferulic acid in combination with other potential natural antioxidants.

## 1. Introduction

Dehydration is a food preservation process, which consists of reducing the moisture levels of foods to prevent spoilage and preserve their quality [1]. The United States Department of Agriculture (USDA) defines dried meat as meat that has been preserved by removing moisture, usually through methods such as air drying, sun drying, smoking, or the use of a dehydrator [2]. This may be made from different types of meat (beef, pork, etc.) and may not necessarily include the seasoning or curing processes, which are typical in jerky production [3]. Dried meat and beef jerky are often confused, but they have distinct characteristics and production methods. Beef jerky is made from beef seasoned and cured before drying. If a seasoning process is included, brine usually contains salt, sugar, and nitrites. The production of dried meat may not necessarily include the seasoning or curing processes, which are typical in jerky production and, if the curing process is included, the brine does not contain sugar.

Dehydration and curing processes not only improve the taste but also extend the shelf life of meat by inhibiting microbial growth [2]. One of the first non-microbial causes of spoilage in dehydrated meat is oxidation [4]. For this reason, the meat industry uses nitrites to give dried meat its unique characteristics of color and flavor, although they are also used to inhibit the oxidation of lipids and proteins and limit the growth of microorganisms [5]. However, they are known as carcinogens, especially if consumed consistently or in excess [5]. In fact, the International Agency for Research on Cancer (IARC) has shown that nitrite consumption from processed beef may lead to colorectal cancer in humans [6]. The lethal oral dosage for humans ranges from 33 to 250 mg of nitrite per kilogram of body mass [6]. Meanwhile, prospective studies in the United States of America and the European Union, as well as meta-analyses of epidemiological data, have found that eating processed meat over a long time is linked to higher mortality rates for colorectal cancer, type 2 diabetes, and heart disease [7]. Alternative replacements for synthetic antioxidants are natural polyphenolic compounds extracted from plants, such as ferulic acid (FA). FA is the most abundant phenolic compound of the hydroxycinnamate group; it is part of the cell wall of plants, fruits, vegetables, feed, and cereal grains and has a wide range of health-related effects, including its antioxidant activity [8,9]. FA not only eliminates free radicals but also inhibits the generation of enzymes that generate free radicals and enhances the activity of enzymes responsible for their removal [10].

In parallel, protein consumption trends vary significantly from region to region due to cultural patterns, economic factors, and eating habits. Thus, in Western countries, there is a preference for the consumption of plant-based proteins, while in other regions traditional animal protein is still prioritized. However, regardless of the protein source, a favorable proportion of consumers seek out foods made with natural ingredients [11]. Therefore, the objective of this study was to evaluate the effect of the addition of FA on the physicochemical, antioxidant, antimicrobial, and sensory characteristics of dried meat during its shelf life.

## 2. Materials and Methods

### 2.1. Treatments

The brine was prepared by dissolving 50 g (5%) of salt and 10 g (1%) of sugar in 1 L of water. For the control treatment (CT), no nitrates or ferulic acid were added. For the nitrite treatment (NT), 20 g (2%) of cured salt (6.25% sodium nitrite and 93.75% salt) were added to the brine. This percentage gives a sodium nitrite concentration of 1250 ppm (parts per million). For the ferulic treatments (FTs), two additional percentages were used to substitute cured salt in the brine: 0.05 and 0.1% (FT05 and FT1, respectively). The concentrations of ferulic acid used to elaborate the brine were determined based on preliminary experiments, starting from the reported solubility of FA in pure water (0.4 to 0.5 g/L) [12].

### 2.2. Dried Meat Elaboration

An Angus beef round steak weighing 6.500 kg was meticulously stripped of external fat, frozen, mildly softened for consistent slicing, and sliced to a thickness of 7 mm perpendicular to the muscle fibers. Manufacturing commenced only when all slices had fully thawed (0–3.3 °C). Then, the meat was divided into 12 portions to elaborate three replicates (540 g) of each treatment. To prepare the dried meat, each slice was submerged for 15 min in the corresponding brine and then left to drain for 10 min on a tray at room temperature. Then, the meat was placed in the dehydrator (Nictemaw, FD-08DGP-L, Shenzhen, China) for 4 h at 70 °C. Once the drying process was finished, the meat was left to cool and packed in polystyrene bags. Dried meat was preserved in a dry, dark place at ambient temperature for six months. All analyses were carried out in triplicate. Physicochemical composition, pH, color, and sodium content were determined after 24 h of meat elaboration. Antioxidant activity and oxidation were analyzed each month, from day 1 to 180. Microbiological analyses were conducted on days 1, 60 and 180.

### 2.3. Physicochemical Composition

Physicochemical analysis (moisture, ash, fat, and protein) according to the AOAC showed measurements of 926.08, 935.42, 989.05, and 991.20, respectively [13].

### 2.4. pH

The pH was evaluated using a potentiometer (Thermo Scientific^TM^, Orion^TM^, Versa Star Pro^TM^, Vantaa, Finland), previously calibrated. First, the dried meat was ground in a grinder (Hamilton Beach, 80335RV, Shenzhen, China). Subsequently, 5 g of meat was combined with 45 mL of distilled water and agitated for two minutes until homogeneity was achieved. The pH electrode was submerged in the mixture until the monitor displayed a stable pH reading. This measurement was conducted in triplicate.

### 2.5. Sodium Determination

The sodium determination was conducted according to the Mexican Official Standard NOM-F-150 S-1981 (Food for Humans—Determination of sodium chloride in brines) [14].

### 2.6. Color

Color was quantified using a CR-410 colorimeter (Konica Minolta^®^, Tokyo, Japan) employing the CIELAB system for L*, a*, and b* measurements. L* serves as an indicator of lightness, ranging from black to white, while a* values extend from green (negative) to red (positive), and b* values range from blue (negative) to yellow (positive). The assessments were conducted on the surface of desiccated meat in triplicate, utilizing the L*, a*, and b* data to compute the color difference (∆E). The ΔE of the treatments was calculated using as a reference the average of the L*, a*, and b* parameter readings for the control treatment and using the following formula [15]:(1)$$\Delta E=\sqrt{{\left(L_{s}-L_{c}\right)}^{2}+{\left(ah_{s}-ah_{c}\right)}^{2}+{\left(bh_{s}-bh_{c}\right)}^{2}}$$
where ΔE is the color difference; L_c_, ah_c_, and bh_c_ are the control values; and L_s_, ah_s_, and bh_s_ are the treatment values.

### 2.7. Antioxidant Activity

Antioxidant activity (AA) was investigated by the ABTS, DPPH, and FRAP methodologies. Each assessment was conducted in triplicate as outlined below.

The AA, via the ABTS method, was assessed as described by Thaipong et al. (2006) [14] and expressed as mg Trolox equivalent (mg TE/100 g), utilizing a Trolox calibration curve (y = −10.467x + 1.0323, R^2^ = 0.9758).

The AA, via the DPPH (2,2-Diphenyl-1-Picrylhydrazyl) technique, was assessed as described by Thaipong et al., (2006) [16], and the results were reported as mg Trolox equivalent (mg TE/mL) using a standard curve (y = −8.3715x + 1.0209, R^2^ = 0.960).

AA by the FRAP method was conducted following Sigma-Aldrich commercial kit instructions, and it was reported as mM ferrous equivalent (mM Fe^2+^ equivalents) using a standard curve (y = 0.0624971 + 0.0328649, R^2^ = 0.992).

### 2.8. Lipid Oxidation

Lipid oxidation was determined by the quantification of thio-barbituric acid reactive substances (TBARS) according to Pfalzgraf et al., (1995), adjusting volumes for microplate reading [17]. TBARS result values were expressed as milligrams (mg) of malondialdehyde (MDA) per kilogram of dried meat, according to a standard calibration curve (y = 144.2x + 0.0066, R^2^ = 0.9977).

### 2.9. Microbiological Analysis

Enumerations for microbial determinations were made for total aerobic mesophilic (TAC, AOAC 990.12), coliforms (AOAC 991.14), and mold and yeast (AOAC 997.02) counts. Colony-forming units (CFU) were counted on plates, with CFU between 10 and 200, and the results were transformed from CFU/g to Log_10_ CFU/g [13].

### 2.10. Sensory Evaluation

The sensory evaluation was conducted using an untrained panel of 59 participants. Meilgaard et al., (2007) state that, for laboratory testing, a panel of 25 to 50 people per product is recommended [18]. Panelists aged 18 to 60 years sensorially evaluated the dried meat (NT and FT1) using a simple difference test. The objective of this test is to determine if there is a sensory difference between two products. Samples were designated with a three-digit code and presented to the panelists in a random order. Panelists were directed to cleanse their palates with water between samples. The NT and FT1 treatments were selected because the first includes dried meat prepared with nitrites, which is how most consumers consume it, and the second because it was the treatment that showed the greatest antioxidant activity. In addition, panelists were asked about their frequency of consumption of dried meat.

### 2.11. Statistical Analysis

A completely randomized one-way design was used. The response variables were analyzed using one-way ANOVA with the General Linear Model (SAS v9.4). Tukey’s test (α = 0.05) was used for post hoc comparisons. Regarding the sensory evaluation, a Chi^2^ test was performed, with the degrees of freedom equal to 1 and a α value of 0.05. Correlations among variables were performed with Pearson’s correlation test.

## 3. Results and Discussion

### 3.1. Physicochemical Composition and Correlation Between Variables

The physicochemical results are shown in Table 1. The pH of the samples was 5.5 (*p* > 0.05) for all the treatments. The protein and ash content did not show statistically significant differences (*p* > 0.05) among treatments. Regarding fat percentage, CT (14.77 ± 0.51) presented the lowest content (*p* < 0.05) compared to the other treatments, but which was not statistically significant (*p* > 0.05). Regarding moisture content, CT had the highest value (7.58 ± 0.52), followed by FT1 (6.4 ± 0.21) (*p* < 0.05), and then by NT (5.2 ± 0.30) and FT05 (5.24 ± 0.33), but with no statistical differences between them (*p* > 0.05). NT also had the highest sodium content (*p* < 0.05, 3130.56 mg Na/100 g), and CT (2715.28 mg Na/100 g), TF05 (2507.64 mg Na/100 g), and FT1 (2539.58 mg Na/100 g) did not present statistical differences (*p* > 0.05).

The pH of dried meat is influenced by a combination of intrinsic factors, such as meat composition and post mortem changes, as well as extrinsic factors, like processing methods and storage conditions [1]. The pH found in this study coincides with that reported in beef jerky when hot air-dried and microwave-assisted and ranged from 5.56 to 5.58 [19].

The fat content found in this study was lower than those reported in beef jerky (18.2 ± 0.551) and higher than those reported for chicken (8.23 ± 0.74) and ostrich (4.49 ± 0.14) jerky [20]. As said before, the variability of fat percentage may be due to the breed and age of the animal, as well as the type of muscle used to elaborate the dried meat [20]. On the other hand, the traditional method for making dried meat often involves removing the fat from the meat; however, as this is done manually, it is possible to retain some fat in the piece used. This could explain the differences found in the fat content between treatments in this study.

Regarding moisture content, dried meat should have up to 10% moisture to protect meat quality during storage time; in this study the four treatments were below that value [20]. Studies also show a negative correlation between moisture and fat content in meat; as fat content increases, moisture content tends to decrease, and vice versa [21]. This correlation was observed in CT, NT, FT05, and FT1 treatments (Table 2, Table 3, Table 4 and Table 5). This relationship is particularly evident in various types of meat, including pork and beef. Although this correlation was not present in CT, the NT, FT05, and FT1 treatments had higher fat content and lower moisture content, contrary to CT, which had higher moisture and lower fat content (Table 1). In this regard, research indicates that, when beef samples are cured with acidic solutions, the moisture content tends to be higher compared to control samples cured solely with salt [22]. The brine used in FT1 had a pH of 3.9, and the brine for FT05 had a pH of 4.3. In addition, FA has a carboxylic acid group (-COOH) and a phenolic hydroxyl group (-OH); these functional groups can interact with water through hydrogen bonding when they are protonated [23]. The above occurs when these functional groups are found in solutions at low pH.

The capacity of FA to engage with water at low pH underscores its potential to preserve moisture levels in dehydrated meat formulations, hence enhancing texture and shelf stability. This potential could also be used in the development of meat products where nitrites are used.

Concerning sodium content, NT presented the highest; this could be attributed to the addition of the cured salt during the curing process, since this is composed of sodium [24]. Likewise, this treatment also showed the lowest moisture content, along with FT05. The presence of NaCl creates an osmotic environment that causes water to leave the cells by osmosis. This process leads to cellular dehydration and therefore to moisture loss. Meat curing is conventionally defined as the process of incorporating ingredients like salt, nitrite, and sodium nitrate into fresh meat slices to extract moisture and diminish the water activity of the tissues, hence preventing rotting and achieving the desired coloration [5,24,25,26].

### 3.2. Color

The values of color are shown in Table 6. In terms of L*, the CT had the lowest value (*p* < 0.05, 28.99 ± 0.45). NT (35.80 ± 0.16) and FT05 (35.20 ± 0.61) had the greatest values (*p* < 0.05), although those for FT05 were not statistically different (*p* > 0.05) from FT1 (34.54 ± 0.44).

Regarding a*, all treatments were positive, indicating a tendency to red. The greatest a* values (*p* < 0.05) were for NT (11.05 ± 0.09) and FT05 (10.50 ± 0.39), although these did not present statistical differences (*p* > 0.05) between them. These were followed by FT1 (9.58 ± 0.57) and CT (6.23 ± 0.18), which had the lowest a* value (*p* < 0.05).

In terms of b*, all treatments presented positive values, indicating a tendency to yellow. FT05 (12.56 ± 0.11) had the highest value (*p* < 0.05), followed by CT (9.75 ± 0.09) and FT1 (9.97 ± 0.37), which did not have statistically significant differences between them (*p* > 0.05). Finally, CT presented the lowest b* value (4.57 ± 0.47, *p* < 0.05).

To obtain the ∆E, the NT values were considered as the control, since nitrites are added to cured meat to develop its characteristic color and flavor [6]. Values of ΔE between 0 and 3.0 indicate that the color change is not noticeable to the human eye [27]. FT1 presented a value under 3.0 (2.98 ± 0.68), and FT05 had a value close to 3.0 (3.01 ± 0.26), which was not different from that of FT1 (*p* > 0.05).

The color of dried meat is an important parameter that affects consumer perception and acceptability. Numerous studies have investigated methods for assessing and comprehending the color attributes of beef, especially regarding jerky production. The ideal color for beef products, including dried meat, is a vibrant cherry red. Deviations towards brownish tones are often associated with quality loss, spoilage, or improper processing conditions [28]. Similarly, a* (10–12), L* (32–33), and b* (9–10) values were reported in dried meat salted by immersion in mixed solutions of NaCl and KCl [29]. However, lower a*, b*, and L* values were reported in beef jerky cured in a solution consisting of 11.5% salt, 3.0% sugar, 3.9% starch syrup, 0.2% black pepper, 0.024% sodium nitrite, and 9.0% salt-water (based on raw meat weight) [30]. These differences could be due to the cured composition solutions.

Meanwhile, a study reported no significant correlation between muscle color and total fat, protein, dry matter, and moisture content in beef meat [31].

Color changes during drying are caused by oxidation, changes in meat surface structure, and non-enzymatic browning reactions [27,32,33]. In processed meats, myoglobin’s (Mb) chemical structure presents noticeable differences depending on the type of packaging used and the process applied. Cooking leads to variations in the color of meat products, from bright cherry-red in bloomed fresh meat to dull brown in cooked meat [34]. The resistance to heat-induced denaturation of Mb is in the following order: deoxy-myoglobin (DeoxyMb) (purple-red) > oxymyoglobin (OxyMb) (red) > metmyoglobin (MetMb) (brown) [35]. In the presence of oxygen, Mb is oxidized to oxymyoglobin and shows a bright pink–red color [31].

Color alterations during drying result from oxidation, modifications in the meat’s surface structure, and non-enzymatic browning reactions [27,32,33]. The chemical structure of myoglobin (Mb) in processed meats exhibits significant variations based on the packaging type and processing methods employed. Cooking results in color differences in meat products, ranging from vibrant cherry-red in blooming fresh meat to dull brown in cooked meat [34]. The thermal stability against denaturation of myoglobin (Mb) follows this hierarchy: deoxy-myoglobin (DeoxyMb) (purple-red) > oxymyoglobin (OxyMb) (red) > metmyoglobin (MetMb) (brown) [35]. In the presence of oxygen, myoglobin is oxidized to oxymyoglobin, exhibiting a vibrant pink–red hue [31].

The development of brown MetMb while cooking arises from the oxidation of the three ferrous forms to a ferric state, contributing to meat coloring. However, a decrease in OxyMb concentration was observed with increasing acidity [31]. As mentioned above, the brine for FT1 had a lower pH than the brines for FT05 and NT, which could explain why FT1 presented a lower a* value compared to FT05 and NT and a lower L* value than NT.

FA is a phenolic compound known for its antioxidant capabilities. In the context of meat products, it helps to mitigate oxidation, which can adversely affect color stability, which is due to the formation of metmyoglobin and other pigments that can darken the meat. In terms of consumer preferences, the above implies that regular consumers of dried meat will continue to buy this product because they will find it to be the same as that which they were used to eating.

### 3.3. Antioxidant Activity (AA)

#### 3.3.1. AA by ABTS Method

The ABTS assay quantifies the ability of antioxidants to neutralize the ABTS cation radical, a blue–green chromophore exhibiting peak absorption at 734 nm, whose strength diminishes in the presence of antioxidants [36]. Differences were found (*p* < 0.05) between treatments (*p* < 0.05) and over time (Figure 1). The highest AA was observed at day 1 (*p* < 0.05) in FT05 and FT1. After 180 days of shelf life, FT1 (20.53 mg TE/100 g) and FT05 (17.69 mg TE/100 g) presented the highest AA (*p* < 0.05).

#### 3.3.2. AA by DPPH Method

The 2,2-diphenyl-1-picrylhydrazyl radical (DPPH) scavenging method is among the most widely used and offers the first approach to assess total AA. The technique relies on the transfer of electrons from antioxidants to neutralize the DPPH radical. The response is characterized by a color shift observed at 517 nm, with the discoloration serving as a sign of antioxidant efficacy [36]. The results of AA for the DPPH method are shown in Figure 2. In general, dried meat treatments with FA had consistently higher AA as measured by the ABTS, DPPH, and FRAP assays. This tendency remained across different treatments and over time.

#### 3.3.3. AA by FRAP Method

This method assesses the capacity of antioxidants to convert ferric ion (Fe^3+^) into ferrous ion (Fe^2+^). The AA for the FRAP method is shown in Figure 3. All treatments showed significant differences (*p* < 0.05). FT1 showed higher AA, followed by FT05, NT, and CT.

The predominant techniques for evaluating antioxidants in meat systems include TBARS, TEAC, ORAC, DPPH, FRAP, and the Folin–Ciocalteu assay. Among these methods, the most widely used to evaluate the antioxidant activity in meat is TBARS, which is based on the estimation of malondialdehyde (MDA) content. This is because MDA is one of the final products of lipid oxidation [37]. However, this technique does not estimate the residual antioxidant activity of foods. For this reason, the antioxidant activity of the treatments was evaluated by the DPPH, ABTS, and FRAP methods. To date, various studies have evaluated the antioxidant activity of natural antioxidants derived from vegetables and fruits on the quality and oxidation of beef and beef products [38,39] and in dry fermented meat products [40]. However, no studies were found that evaluate the addition of FA to dried meat. In this study, compared to CT, the addition of FA at 0.1% (FT1) increased the antioxidant activity of dried meat in a range from 78.65 to 321.24% (depending on the technique used to evaluate AA). In FT05, it increased from 72.64 to 262.41%, and in NT it increased from 26.31 to 190%. The antioxidant potential of dried meat increased by over 600% when elaborated with 15% of puréed raisins [41]. In another study, balloon flower root extract (BFE), Japanese apricot extract (JAE), and grape extract (GE) were added (0.05% (*v*/*v*) in the formulation of dried meat); although all extracts presented AA, none of these presented higher AA than that presented by the positive control containing butylated hydroxy anisole (BHA) [42].

The potent antioxidant properties of FA primarily encompass the suppression of reactive oxygen species (ROS) and reactive nitrogen species (RNS) generation, alongside the neutralization of free radicals, so impeding a cascade of processes that produce free radicals [41]. Furthermore, FA can serve as a hydrogen donor, supplying atoms to free radicals to safeguard fatty acids from oxidative damage. Moreover, ferulic acid possesses the ability to bind Cu(II) or Fe(II) metal ions, hence inhibiting the generation of harmful hydroxyl radicals. Moreover, it functions as an inhibitor of enzymes that facilitate the production of free radicals and augments the activity of scavengers [41]. This confers a superiority over alternative natural antioxidants.

Eating foods that possess antioxidant activity can provide a variety of health benefits for consumers. Antioxidants are compounds that help neutralize free radicals, which are unstable molecules that can cause oxidative stress, leading to cellular damage and various diseases. Antioxidant-rich foods help to reduce oxidative stress and inflammation and may lower the risk of certain types of cancer, improve skin, brain, and heart health, and support the immune system. In addition, natural antioxidants, derived from plant sources, are often more bioavailable, which means they are more easily absorbed and utilized by the body, have superior safety profile, and present additional health benefits [43,44,45].

The addition of natural antioxidants to meat formulations helps reduce lipid oxidation and improve health benefits by lowering harmful oxidative products. This is particularly relevant as consumers increasingly seek healthier meat options [46,47].

In terms of marketing this snack, the results could help increase the number of consumers if the labeling highlights the use of natural ingredients in its preparation, as well as the fact that it contains less sodium than the original product, is a source of protein and has antioxidant activity.

### 3.4. Lipid Oxidation

The predominant technique for assessing oxidation involves quantifying the oxidized malondialdehyde (MDA) in meat and meat products [48]. MDA is a primary product of lipid oxidation and is linked to the formation of unpleasant scents in meat. [49]. The oxidation results are shown in Figure 4. The four treatments showed similar behavior during their shelf life, where FT05 and FT1 presented the lowest values and CT and NT the largest (*p* < 0.05).

All values obtained in this study were below the threshold of perceived rancidity (2.0 mg MDA/kg) and remained below the acceptability threshold of 1 mg MDA/kg [50]. Zioud et al., (2023) reported lower values for lamb meat powdered and spiced with garlic, coriander, salt, and paprika drying by convective (0.35 mg MDA/kg) or sun-drying (0.93 mg MDA/kg) methods [51]. Similarly, Mediani et al. (2022) reported lower values in sun-dried (0.77 mg MDA/kg) and air-dried (0.68 mg MDA/kg) meat [1]. Higher values than those found in this study were also reported in beef jerky cured for 24 h with salt (2.94 ± 0.11 mg MDA/kg), soy sauce (3.85 ± 0.07 mg MDA/kg), red pepper paste (1.59 ± 0.20 mg MDA/kg), and soybean paste (2.46 ± 0.02 mg MDA/kg) [22].

Chen et al., (2004) observed that there is a proportional increase of lipid oxidation related to the presence of NaCl, which acts as pro-oxidant [52]. This could explain the higher values of MDA found in NT, since nitrites are composed of residual sodium salts.

It is important to highlight that one limitation of using FA in food formulation is its low solubility in water. However, in the treatments with FA, it was added at 0.05% (FT05) and 0.1% (FT1), and at these concentrations there were no solubility problems. This may be due to the presence of salts, such as sodium chloride (NaCl). Although the dissociation constants (pKa, 4.5 for the carboxylic acid group and 9.0 for the phenolic hydroxyl group) of ferulic acid are above the pH of brines added with ferulic acid, in the presence of ions in solution protective charges may form, affecting the ease with which protons are released from ferulic acid [12]. Higher ionic strength often results in lower pKa values due to reduced electrostatic repulsion between charged species [12]. Therefore, if the carboxyl group is charged (deprotonated) and the phenolic hydroxyl group is uncharged (protonated), they can interact with sodium ions (Na⁺) through ionic interactions or hydrogen bonds, thus increasing their solubility.

Finally, despite the lower percentage of FA added in the treatments in this study, these presented lower MDA values. This reflects its potential as an antioxidant agent, although it is worth mentioning that no studies were found in which FA was used in dried meat.

### 3.5. Microbiological Analysis

Total coliforms and mold and yeast counts were below the detection limit (<2 Log_10_ CFU/g) on any sampled day. Regarding total aerobic mesophiles counts at day 1, CT and NT (4.86 and 4.75 Log_10_ CFU/g) did not show statistical differences between them (*p* > 0.05); however, these were statistically different from TF05 and TF1 (4.30 and 4.00 Log_10_ CFU/g, respectively, *p* < 0.05). Then, at days 60 and 180, counts were below the detection limit (<2 Log_10_ CFU/g). FA treatments more effectively inhibited the growth of aerobic mesophiles than NT [53]. In addition, spoilage microorganisms (gram-positive and -negative bacteria, molds, and yeasts) are sensitive to hydroxycinnamic acid derivatives, such as caffeic, ferulic, and p-coumaric acids [54]. In another study, FA not only suppressed rotting organisms but also pathogens, including *Escherichia coli* and *Listeria monocytogenes* [55]. FA functions as a significant food additive owing to its antimicrobial properties, which stem from the inhibition of arylamine N-acetyltransferase, an enzyme that facilitates arylamine acetylation in bacteria, resulting in irreversible alterations to cell morphology and membrane characteristics, including charge and permeability, both intra- and extracellularly [8]. The microbiological attributes of dried meat are often characterized by the standard total plate count. (<5 Log_10_ CFU/g), with values similar to the results (4.0 to 4.86 Log_10_ CFU/g) in this study [1] (Table 7).

### 3.6. Sensory Evaluation

From the 59 panelists, 45.76% were women and 54.23% were men. They were asked about their frequency of consumption of dried meat; 16.9% ate it once a week, 40.6% 2–3 times a month, 40.6% once every 2 months, and 1.6% did not eat it. Panelists did not detect differences among the treatments (X^2^ = 2.10, *p* > 0.05); consequently, the addition of FA did not present changes in consumer perception. Nowadays, the use of natural additives instead of synthetic additives is being widely accepted; likewise, the development of new meat products with improved nutritional profiles has increased in the last decade as consumers demand meat products without artificial preservatives, minimally processed, and with a longer shelf life [5].

## 4. Conclusions

This study investigated the effect of FA on the shelf life of dried meat. First, it is important to emphasize that the brines that contain FA were successfully prepared and utilized in the production of dried meat, without presenting solubility problems. Treatment with FA presented higher antioxidant activity and lower lipid oxidation and total aerobic mesophile counts. The physicochemical and color characteristics of dried meat produced with FA were found to be comparable to those of the dried meat elaborated with nitrites. The change from nitrites to ferulic acid was not sensorially detected by panelists. Furthermore, NT had the highest sodium content. This suggests that the utilization of FA to produce dried meat does not negatively impact the principal characteristics of this meat snack. Therefore, it is recommended to be used at a concentration of 0.1%; at this concentration, the antioxidant activity of dried meat is greater, and the oxidation is below the threshold of perceived rancidity. Marketing this snack could enhance consumer engagement if the labeling emphasizes the use of natural ingredients, the reduced sodium content compared to the original product, its protein source, and its antioxidant properties.

## Figures and Tables

**Figure 1 foods-14-00708-f001:**
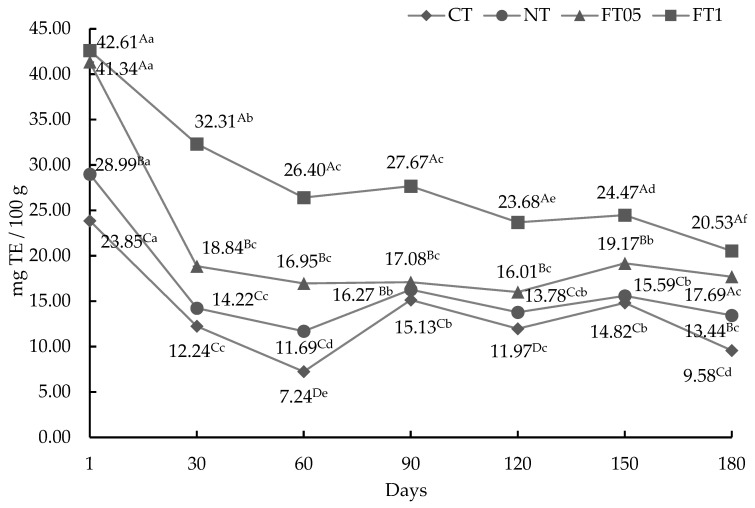
Antioxidant activity by ABTS method for dried meat (mean ± standard deviation) with and without addition of nitrites or ferulic acid (0.1 and 0.05%). TE, Trolox equivalent; CT, dried meat without nitrites or ferulic acid; NT, dried meat with 2% of nitrites; FT05, dried meat with 0.05% of ferulic acid; FT1, dried meat with 0.1% of ferulic acid. ^A,B,C,D^ = uppercase superscripts indicate significant statistical differences between days in the same treatment (*p* < 0.05). ^a–f^ = lowercase superscripts indicate significant statistical differences between treatments on the same day (*p* < 0.05).

**Figure 2 foods-14-00708-f002:**
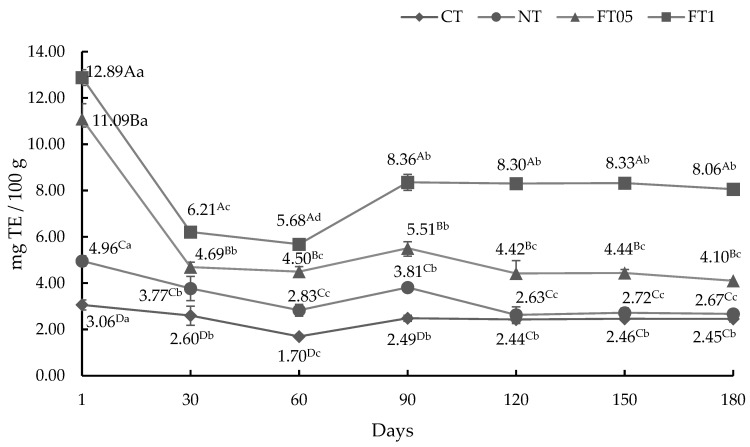
Antioxidant activity for DPPH method of dried meat (mean ± standard deviation) with and without addition of nitrites or ferulic acid (0.1 and 0.05%). TE, Trolox equivalent; CT, dried meat without nitrites or ferulic acid; NT, dried meat with 2% of nitrites; FT05, dried meat with 0.05% of ferulic acid; FT1, dried meat with 0.1% of ferulic acid ^A–D^ = uppercase superscripts indicate significant statistical differences between days in the same treatment (*p* < 0.05). ^a–d^ = lowercase superscripts indicate significant statistical differences between treatments on the same day (*p* < 0.05).

**Figure 3 foods-14-00708-f003:**
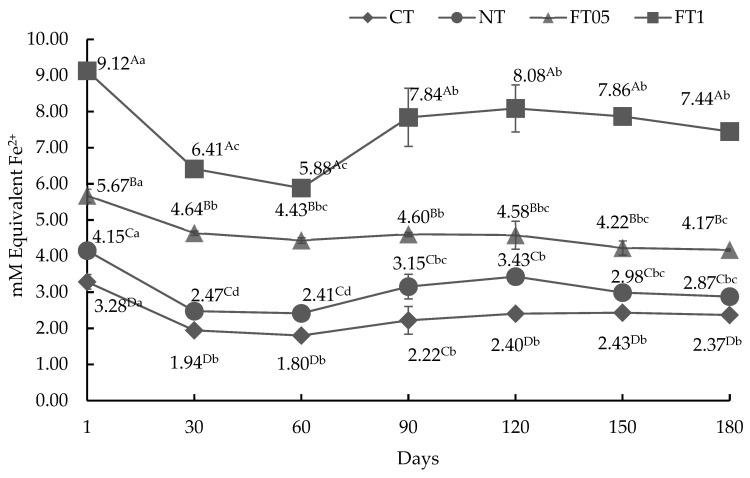
Antioxidant activity for FRAP method of dried meat (mean ± standard deviation) with and without addition of nitrites or ferulic acid (0.1 and 0.05%). CT, dried meat without nitrites or ferulic acid; NT, dried meat with 2% of nitrites; FT05, dried meat with 0.05% of ferulic acid; FT1, dried meat with 0.1% of ferulic acid. ^A–D^ = uppercase superscripts indicate significant statistical differences between days in the same treatment (*p* < 0.05). ^a–d^ = lowercase superscripts indicate significant statistical differences between treatments on the same day (*p* < 0.05).

**Figure 4 foods-14-00708-f004:**
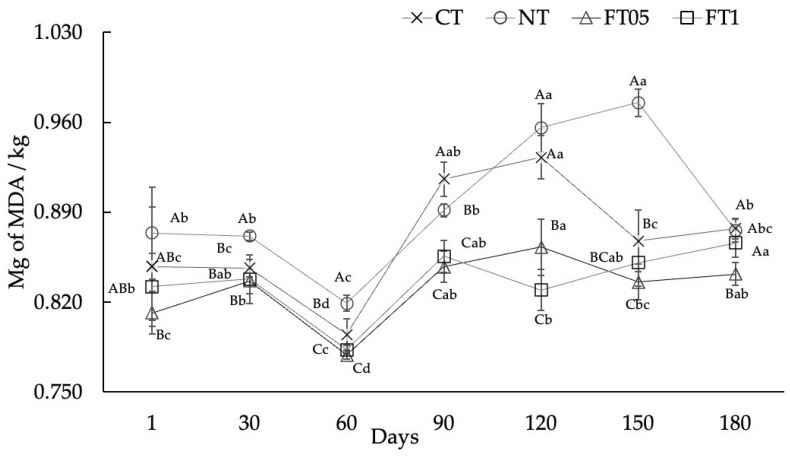
Oxidation of dried meat (mean ± standard deviation) with and without addition of nitrites or ferulic acid (0.1 and 0.05%). MDA, malondialdehyde; CT, dried meat without nitrites or ferulic acid; NT, dried meat with 2% of nitrites; FT05, dried meat with 0.05% of ferulic acid; FT1, dried meat with 0.1% of ferulic acid ^A–C^ = uppercase superscripts indicate significant statistical differences between treatments on the same day (*p* < 0.05). ^a–d^ = lowercase superscripts indicate significant statistical differences between days on the same treatment (*p* < 0.05).

**Table 1 foods-14-00708-t001:** Physicochemical composition (%) of dried meat (mean ± standard deviation).

Variables	Treatments			
	CT	NT	FT05	FT1
Protein	64.05 ± 3.08 ^a^	64.05 ± 1.63 ^a^	64.84 ± 1.56 ^a^	62.82 ±1.16 ^a^
Fat	14.77 ± 0.51 ^b^	15.77 ± 0.50 ^a^	15.42 ± 0.28 ^ab^	15.75 ± 0.09 ^a^
Ashes	13.40 ± 0.90 ^a^	12.38 ± 1.01 ^a^	14.25 ± 1.26 ^a^	14.49 ± 1.26 ^a^
Moisture	7.58 ± 0.52 ^a^	5.20 ± 0.30 ^c^	5.24 ± 0.33 ^c^	6.40 ± 0.21 ^b^
pH	5.50 ± 0.00 ^a^	5.50 ± 0.00 ^a^	5.50 ± 0.00 ^a^	5.50 ± 0.00 ^a^
Sodium (mg/100 g)	2715.28 ± 22.11 ^a^	3130.56 ± 25.80 ^b^	2507.64 ± 30.83 ^a^	2539.58 ± 29.25 ^a^

CT, dried meat without nitrites or ferulic acid; NT, dried meat with nitrites; FT05, dried meat with 0.05% of ferulic acid; FT1, dried meat with 0.1% of ferulic acid. ^a,b,c^ = Different literals between columns denote significant differences (*p* < 0.05) among treatments.

**Table 2 foods-14-00708-t002:** Correlation between physicochemical composition and color variables of CT.

	Moisture	Ashes	Fat	Protein	L	a	b	Sodium
Moisture	1	−0.375	0.879	−0.966	−0.817	0.018	1.000 *	−0.484
*p*-value		0.755	0.316	0.168	0.391	0.988	0.018	0.678
Ashes		1	−0.772	0.121	0.840	0.920	−0.401	0.993
*p*-value			0.439	0.923	0.365	0.256	0.737	0.077
Fat			1	−0.725	−0.993	−0.461	0.892	−0.843
*p*-value				0.484	0.074	0.695	0.298	0.362
Protein				1	0.639	−0.278	−0.958	0.240
*p*-value					0.558	0.821	0.186	0.846
L					1	0.561	−0.833	0.900
*p*-value						0.621	0.373	0.287
a						1	−0.010	0.866
*p*-value							0.993	0.333
b							1	−0.509
*p*-value								0.660
Sodium								1

CT, dried meat without nitrites or ferulic acid. *p*-value = Level of significance of the correlation. * = The correlation is significant at the 0.05 level (two-tailed). L* = lightness, lighter (+) and darker (−); a* = green (−) and red (+); b* = blue (−) and yellow (+).

**Table 3 foods-14-00708-t003:** Correlation between physicochemical composition and color variables of NT.

	Moisture	Ashes	Fat	Protein	L	a	b	Sodium
Moisture	1	1.000 **	−1.000 **	−1.000 **	1.000 **	1.000 **	−0.866	0.000
*p*-value		0.000	0.004	0.004	0.000	0.000	0.333	1.000
Ashes		1	−1.000 **	−1.000 **	1.000 **	1.000 **	−0.866	0.000
*p*-value			0.004	0.004	0.000	0.000	0.333	1.000
Fat			1	1.000 **	−1.000 **	−1.000 **	0.869	0.007
*p*-value				0.009	0.004	0.004	0.329	0.996
Protein				1	−1.000 **	−1.000 **	0.863	−0.006
*p*-value					0.004	0.004	0.337	0.996
L					1	1.000 **	−0.866	0.000
*p*-value						0.000	0.333	1.000
a						1	−0.866	0.000
*p*-value							0.333	1.000
b							1	0.500
*p*-value								0.667
Sodium								1

NT, dried meat with nitrites. *p*-value = Level of significance of the correlation. ** = The correlation is significant at the 0.01 level (two-tailed). L* = lightness, lighter (+) and darker (−); a* = green (−) and red (+); b* = blue (−) and yellow (+).

**Table 4 foods-14-00708-t004:** Correlation between physicochemical composition and color variables of FT05.

	Moisture	Ashes	Fat	Protein	L	a	b	Sodium
Moisture	1	1.000 **	−1.000 **	−1.000 **	1.000 **	1.000 **	−0.866	0.866
*p*-value		0.000	0.004	0.004	0.000	0.000	0.333	0.333
Ashes		1	−1.000 **	−1.000 **	1.000 **	1.000 **	−0.866	0.866
*p*-value			0.004	0.004	0.000	0.000	0.333	0.333
Fat			1	1.000 **	−1.000 **	−1.000 **	0.869	−0.869
*p*-value				0.000	0.004	0.004	0.329	0.329
Protein				1	−1.000 **	−1.000 **	0.869	−0.869
*p*-value					0.004	0.004	0.329	0.329
L					1	1.000 **	−0.866	0.866
*p*-value						0.000	0.333	0.333
a						1	−0.866	0.866
*p*-value							0.333	0.333
b							1	−1.000 **
*p*-value								0.000
Sodium								1

FT05, dried meat with 0.05% of ferulic acid. *p*-value = Level of significance of the correlation. ** = The correlation is significant at the 0.01 level (two-tailed). L* = lightness, lighter (+) and darker (−); a* = green (−) and red (+); b* = blue (−) and yellow (+).

**Table 5 foods-14-00708-t005:** Correlation between physicochemical composition and color variables of FT1.

	Moisture	Ashes	Fat	Protein	L	a	b	Sodium
Moisture	1	1.000 **	−1.000 **	−1.000 **	1.000 **	1.000 **	−0.866	0.866
*p*-value		0.000	0.004	0.004	0.000	0.000	0.333	0.333
Ashes		1	−1.000 **	−1.000 **	1.000 **	1.000 **	−0.866	0.866
*p*-value			0.004	0.004	0.000	0.000	0.333	0.333
Fat			1	1.000 **	−1.000 **	−1.000 **	0.869	−0.869
*p*-value				0.009	0.004	0.004	0.329	0.329
Protein				1	−1.000 **	−1.000 **	0.863	−0.863
*p*-value					0.004	0.004	0.337	0.337
L					1	1.000 **	−0.866	0.866
*p*-value						0.000	0.333	0.333
a						1	−0.866	0.866
*p*-value							0.333	0.333
b							1	−1.000 **
*p*-value								0.000
Sodium								1

FT1, dried meat with 0.1% of ferulic acid. *p*-value = Level of significance of the correlation. ** = The correlation is significant at the 0.01 level (two-tailed). L* = lightness, lighter (+) and darker (−); a* = green (−) and red (+); *b = blue (−) and yellow (+).

**Table 6 foods-14-00708-t006:** Color values of dried meat (mean ± standard deviation).

Color	Treatments
CT	NT	FT05	FT1
L*	28.99 ± 0.45 ^c^	35.80 ± 0.16 ^a^	35.20 ± 0.61 ^ab^	34.54 ± 0.44 ^b^
a*	6.23 ± 0.18 ^c^	11.05 ± 0.09 ^a^	10.50 ± 0.39 ^a^	9.58 ± 0.57 ^b^
b*	4.57 ± 0.47 ^c^	9.75 ± 0.09 ^b^	12.56 ± 0.11 ^a^	9.97 ± 0.37 ^b^
∆E	10.01 ± 0.31 ^a^	_______	3.01 ± 0.26 ^b^	2.98 ± 0.68 ^b^
	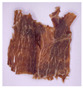	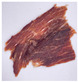	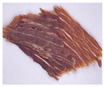	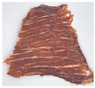

CT, dried meat without nitrites or ferulic acid; NT, dried meat with nitrites; FT05, dried meat with 0.05% of ferulic acid; FT1, dried meat with 0.1% of ferulic acid. ^a,b,c^ Different literals between columns denote significant differences (*p* < 0.05) between treatments. L* = lightness, lighter (+) and darker (−); a* = green (−) and red (+); b* = blue (−) and yellow (+); ΔE = color difference. ^a,b,c^ Different literals between rows show a significant difference (*p* < 0.05) between treatments.

**Table 7 foods-14-00708-t007:** Total aerobic mesophile counts (Log_10_ CFU/g) for dried meat (mean ± standard deviation).

Days	Treatments			
	CT	NT	FT05	FT1
1	4.86 ± 0.03 ^a^	4.75 ± 0.17 ^a^	4.30 ± 0.11 ^b^	4.00 ± 0.01 ^b^
60	<2	<2	<2	<2
180	<2	<2	<2	<2

CT, dried meat without nitrites or ferulic acid; NT, dried meat with nitrites; FT05, dried meat with 0.05% of ferulic acid; FT1, dried meat with 0.1% of ferulic acid. ^a,b^ = Different literals between columns denote significant differences (*p* < 0.05) among treatments.

## Data Availability

The original contributions presented in the study are included in the article, further inquiries can be directed to the corresponding author.

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
