# Peer review of "Antioxidant and Antimicrobial Activity of Ferulic Acid Added to Dried Meat: Shelf-Life Evaluation"

_foods, 2025, doi:10.3390/foods14040708_

Round 1
Reviewer 1 Report
Comments and Suggestions for Authors
The authors submitted the manuscript entitled “Antioxidant and Antimicrobial Activity of Ferulic Acid Added to Dried Meat: Shelf-Life Evaluation” and suggested the improvement of the oxidative stability of the product after the incorporation of ferulic acid.
Analysing the experimental design, I feel that some parameters are missing. For example, why weren't some physicochemical parameters such as humidity tested over time? Why weren't pathogens enumerated/detected in the prototypes developed? How did the authors ensure that the products were safe enough to carry out a sensory analysis (the information is missing, but the authors imply that the samples were ingested). There are several questions that the authors should answer and whose information should be mentioned in the manuscript.
I would advise the authors to proofread the English, as there are some grammatical and typographical errors. The text should also be improved in general, as it sometimes seems more like a report than a scientific article. The discussion must be improved with other studies published in the area.
More suggestions for improvement:
Line 14: What nitrite was used and in what quantity?
Line 82 and 83: “g and grams”. Please uniformize.
Line 98: “packed in 98 polystyrene bags”. No modified atmosphere was used?
Line 103: “after 24 hours of storage” Why this sampling point? Why didn't the authors test over the shelf-life?
Line 108: “Then, 5 g of meat were taken and mixed with 45 mL of distilled water”. Shouldn't the pH have been measured with a solid electrode? Don't the authors think that this dilution affects the value obtained? And which samples were measured? At all points of the shelf-life?
Line 141: Authors should check the journal's guidelines regarding the formatting of references that appear throughout the text.
Line 158: The centrifugation conditions should be mentioned.
Line 159: “TBA reagent” – Authors should write the name in full and only then use the acronym.
Line 167: Was the enumeration (or detection) of any pathogens tested? Did it not make sense for the authors to perform challenge tests on the tested prototypes? This should be mentioned.
Line 169: “plate count enriched agar” – Do the authors mean Plate Count Agar? This medium is not enriched.
Line 179: Molds and yeasts were incubated at 25 ºC for how long?
Line 182: Which attributes were tested? They should be mentioned.
Table 1: The results presented must have the same number of decimal places.
Line 350: “Figure 1” instead of “Graph 1”. Please change to the others below.
Line 443: Please eliminate “p”.
Line 462-469: References are needed to support the various sentences.
Line 483: “TC and TN (4.6 and 4.75 Log10 CFU/mg)”. The values do not coincide with those shown in the table.
Line 484: I can understand that the statistical analysis gave significant differences between the treatments, however, a difference of around 0.5 log cycles is not biologically significant. I think the authors should revise the discussion on this part.
Table 2: What does ND mean? For these samples, the result must be < detection limit of the enumeration technique. How do the authors justify the counts obtained for Total aerobic mesophiles at time zero? And didn't the authors find it strange that these numbers decreased during storage? The authors said it was the action of the compounds, but the same thing happened in the control. A more in-depth discussion is needed.
The conclusions are a little vague, as the authors have only summarised the results obtained. It should be revised.
Author Response
Dear reviewer,
First of all, thank you for your comments. The authors read each one of them and responded to your suggestions.

Reviewer 2 Report
Comments and Suggestions for Authors
The manuscript submitted for review “ Antioxidant and Antimicrobial Activity of Ferulic Acid Added to Dried Meat: Shelf-Life Evaluation” contains the interesting results. I have included comments on the manuscript below:
1. Abstract - please expand the abbreviations used to make the abstract more readable.
2. Table 1. Why is the sodium content expressed in different units than the other components. What is the purpose of presenting the results in this way?
Tables: I personally think that tables with correlations between physicochemical indicators and color parameters are not related to the title of the manuscript and could be omitted.
3. How was the color measured - on the surface after drying, on the cross-section?
4. Do the authors have the results for the drying process yield? They should be unconditionally included because the final yield primarily influences the chemical composition. If the additives used affected the drying yield, then the chemical composition was indirectly affected.
5. Conclusions: lines 525 - 526 The sentences “enhanced its shelf life and sensory properties” and “not sensorially detected by panelists.” are mutually exclusive.
The manuscript should be reviewed again after improvement.
Author Response

(The authors gave the same response as above.)

Reviewer 3 Report
Comments and Suggestions for Authors
The objective of this study was to evaluate the effect of the addition of ferulic acid on the physicochemical, antioxidant, antimicrobial, and sensorial characteristics of dried meat during its shelf life. Although the author proposed a method to improve the oxidative stability of dried meat, this research lacks innovation and in-depth analysis, which limits the contribution of the manuscript in this field. Therefore, I have some recommend for the manuscript.
1. Line 62-70, it is suggested to focus on the application of ferulic acid in food.
2. Line 89, why is the addition ratio of ferulic acid 0.05 and 0.1%.
3. In “2.11. Sensory evaluation” section, the evaluation panel consisted of untrained consumers, so are the results scientifically valid? Is there a personal preference? In addition, what is the criteria for evaluation?
4. In 3.3 section, Fig.1, Fig.2, Fig.3, and Fig.4 already present the data clearly and do not need to be described in detail in this article. It is worth noting that the author should discuss the changing trends in depth. Additionally, some paragraphs are lengthy and information-intensive, which may make it difficult for readers to quickly grasp the key points. It is recommended to make the paragraphs clear and optimize the language to make the content more concise and easier to understand.
5. Improve conclusion section.
Author Response

(The authors gave the same response as above.)

Reviewer 4 Report
Comments and Suggestions for Authors
Dear Authors,
My comments are attached.

The English language needs moderate editing.
Author Response

(The authors gave the same response as above.)

Round 2
Reviewer 2 Report
Comments and Suggestions for Authors
The authors complied to all the comments contained in the first review.
I am satisfied with the form of changes and responses.
Author Response
Dear reviewer,
Thank you for taking the time to review the article and for your comments and suggestions, which have undoubtedly enriched and increased its quality.
Reviewer 3 Report
Comments and Suggestions for Authors
I think author has answered the reviewer’s questions and the manuscript has been improved.
Author Response

(The authors gave the same response as above.)

Reviewer 4 Report
Comments and Suggestions for Authors
Dear Authors,
Thank you for the revised manuscript, which shows significant improvements. However, the manuscript requires further English language editing. Please ensure this is addressed before resubmitting.
Comments on the Quality of English LanguageThe manuscript requires further English language editing and improvement.
Author Response
Estimado revisor,
Gracias por tomarse el tiempo de revisar el artículo y de sus comentarios y sugerencias, que sin duda han enriquecido e incrementado la calidad de este.